# Evaluation of Residual Efficacy of Pyrethrin + Methoprene Aerosol on Two Dermestids: Impact of Particle Size, Species, and Temperature

**DOI:** 10.3390/insects10050142

**Published:** 2019-05-17

**Authors:** Srinivas K. Lanka, Frank H. Arthur, James F. Campbell, Kun Yan Zhu

**Affiliations:** 1Department of Entomology, 123 West Waters Hall, Kansas State University, Manhattan, KS 66506, USA; slanka@ksu.edu (S.K.L.); kzhu@ksu.edu (K.Y.Z.); 2USDA-ARS-Center for Grain and Animal Health Research, 1515 College Avenue, Manhattan, KS 66502, USA; james.campbell@ars.usda.gov

**Keywords:** aerosol, methoprene, pyrethrin, *Trogoderma inclusum*, *Trogoderma variabile*

## Abstract

Residual effects of pyrethrin + methoprene aerosol dispensed at 4 and 16-µm particle sizes and an untreated control, was assessed against late-stage larvae of *Trogoderma inclusum* (LeConte), the larger cabinet beetle, and *T. variabile* (Ballion), the warehouse beetle. Treated arenas were stored at 25, 30, 35, and 40 °C and bioassays were conducted at 1, 3, or 6 weeks post-treatment. Larval development was monitored through adult emergence to compare the efficacy of treatments by using both the percentage of normal adult emergence and a developmental index as dependent variables. There was no overall effect of temperature on residual activity as measured using either adult emergence or developmental index values. Both the 4 and 16-µm particle sizes resulted in reduced adult emergence and low developmental index values compared to untreated controls. The insecticide was more effective on *T*. *variabile* than on *T. inclusum.* The impact of particle size varied between species, both particle sizes reduced adult emergence and developmental index in *T. variabile*, but only the 16-µm particle size resulted in reduction of adult emergence of *T. inclusum*. Furthermore, there was a reduction in activity of methoprene with residual exposure time. The variations in susceptibility of species to methoprene, differences in efficacy of particle sizes, and decrease in residual persistence at smaller particle sizes highlight the need for attaining optimal particle size to improve overall efficacy of aerosol mixtures containing methoprene.

## 1. Introduction

Aerosol insecticides can be vital components in management of stored product insects in mills, processing plants, and food storages. Effective use of aerosols as part of an integrated pest management program can be part of a replacement program for the fumigant methyl bromide in the United States (USA) [1,2]. Pyrethrin, a naturally occurring insecticide, is used in the US either alone or in combination with an insect growth regulator (IGR), either methoprene or pyriproxyfen [3,4,5,6,7,8]. Their combined use can accomplish insect population control by direct initial kill of exposed immatures and adults due to immediate contact toxicity of pyrethrin and delayed control of immature stages through exposure to residues of the IGR on treated surfaces. 

Recent research on aerosols has demonstrated the influence of particle size and target insect species on efficacy of aerosols. Studies on efficacy of different particle sizes of pyrethrin aerosol dispensed on adults of *Tribolium confusum* Jacqueline du Val, the confused flour beetle, exposed on treated concrete arenas revealed that the smaller particle size did not kill the adults, while the largest particle size of 16 µm resulted in a significant mortality [8,9]. In another study by Arthur et al. [7], residual efficacy of pyrethrin + methoprene aerosol dispensed at particle sizes of 2 and 16 µm on concrete arenas was evaluated at selected post-treatment intervals for control of late-stage larvae of four beetle species including *T. confusum*, confused flour beetle, *Lasioderma serricorne* F., cigarette beetle, and two dermestids, *Trogoderma variabile* Ballion, warehouse beetle, and *Dermestes maculatus* De Geer, hide beetle. Efficacy, as measured by adult development of exposed immatures, varied not only with the particle size but also with the target species. The authors found that 2 and 16-µm particle sizes suppressed the development of larvae to the adult stage in *L. serricorne,* neither of the particle sizes could suppress development in *D. maculatus*, while in *T. confusum* and in *T. variabile* only the 16-µm particle size suppressed adult emergence. This study has demonstrated striking variability between late larvae of the two dermestid species viz., *D. maculatus* and *T. variabile* in response to methoprene. 

*Trogoderma inclusum* LeConte, larger cabinet beetle and *T. variabile* are dermestid beetles and are important pests that infest mills, processing plants, and a wide range of grain products. Larvae of these species have a wide host range [10,11,12]. Infestations of these species can lead to contamination of food products by the presence of larvae, masses of cast skins, live or dead insects, and frass [10,11,13]. Both species are short-lived as adults and spend most of their life cycle as larvae [14]. Although previous studies on different measures of control in dermestids indicate that these insects are less susceptible to contact insecticides and cold temperatures relative to *Tribolium* spp. [15], the more recent studies with contact insecticides and IGRs showed that *D. maculatus* and *T. variabile* varied in their developmental responses to the methoprene component of pyrethrin + methoprene aerosol [7]. To our knowledge, no information is available on comparing the efficacy of this insecticide mixture against *T. inclusum* and *T. variabile*. 

Most of the information on the influence of temperature on insecticidal persistence against stored product insects comes from general surface treatments. In a long-term study on the impact of temperature on persistence of methoprene treatments at different temperatures in the range of 20–35 °C, Wijayaratne et al. [16] reported that temperature by itself did not alter residual activity of this insecticide against *T. castaneum* larvae. However, the decay in methoprene activity with passage of time out to 24 weeks was evident as measured by increase in adult emergence. In a study by Arthur [17], the decay in activity of deltamethrin on treated arenas occurred mainly during the summer months, not in the autumn or winter months, when held in empty grain bins. This observation was based on slower knockdown of adults exposed on the arenas that were held on the floor of a grain bin outside compared to arenas held inside a building or inside an environmental chamber [17].

Residual activity of contact insecticides and IGRs can vary with target species, levels of initial deposition of the chemical, and environmental conditions after treatment. Therefore, the objectives of this study were to: (1) determine residual efficacy of pyrethrin + methoprene dispensed at particle sizes of 4 µm and 16 µm, (2) examine potential variability between *T. inclusum* and *T. variabile* in response to the aerosol combination, and (3) examine the impact of temperature on residual persistence. 

## 2. Materials and Methods

The life stages used for this study were three to four-week-old late stage larvae of *T. inclusum* and *T. variabile* from insecticide-susceptible colonies maintained at the USDA-ARS Center for Grain and Animal Health Research (CGAHR), Manhattan, KS. The source for *T. inclusum* colony was outside a commercial Kansas mill in 2012. The colony of *T. variabile* was started >30 years ago and original source is unknown. Both colonies were reared on ground dog food in mason jars (0.95 L) with screen lids. The colonies were maintained in walk-in chambers (30 °C, 60% relative humidity (r.h.) and 16:8 h L:D). One hundred and twenty mixed-sex adults of each species obtained from the lab colonies were released into rearing jars containing ground dog food and adults were allowed to mate and oviposit. To provide a refuge for adults in the food, a small amount of oatmeal with rags of cellulose paper were placed on top of the dog food in the jars. These jars for larval production were maintained at similar conditions as mentioned for lab colonies. 

The exposure arenas used for this study were created by filling the bottom of polystyrene Petri dishes, each measuring 62 cm^2^ in area and 2.0 cm depth, with a ready-mix concrete driveway patching material (Rockite, Heartline Products, Cleveland, OH, USA). The process for preparing these arenas has been described in detail in previous publications [7,9,15,17]. Briefly, this involves mixing of the dry powder with water to make a slurry and pouring the slurry into Petri dishes to a depth of ca 1.25 cm. These arenas were left to dry on a laboratory counter for several days, and the inner walls of arenas were coated with a fluoropolymer resin suspension (Fluon®, Sigma Aldrich, St. Louis, MO, USA) to prevent larvae from crawling up the sides so that they would be restricted to the concrete surface. A total of 720 arenas were prepared for this study. 

Exposure to pyrethrin + methoprene aerosol occurred in a vertical air-flow chamber at MRIGlobal, Kansas City, MO, US. The details on the chamber and the aerosol dispersion system were described previously [8,9]. Briefly, this chamber provides a continuous distribution of aerosol particles for as long as the system is operating. A Spraying Systems (Wheaton, IL, USA) air assist nozzle was used to dispense particles of volume median diameter (VMD) of either 4 µm or 16 µm. An exposure time of 15 min was chosen because a previous study revealed similar particle concentrations between the two particle sizes at this exposure time [7]. The particle size diameter and concentration of aerosol was measured using Malvern Spraytec aerosol particle sizer (Malvern Instruments, Worcestershire, UK).

The aerosol was applied as follows. A synergized formulation of 1% active ingredient (AI) pyrethrin with 5% piperonyl butoxide and containing petroleum distillate as a carrier (Entech Fog 10, Entech Systems, Kenner, LA, USA), and a 33% AI methoprene formulation (Diacon^®^ IGR, Central Life Sciences, Schaumberg, IL, USA) were used. As per the label directions, both Entech Fog 10 and Diacon^®^ IGR were mixed in a ratio of 100: 1 (pyrethrins and methoprene AIs were in the ratio of 2.7: 1 in this mixture). A 60 mL pyrethrin formulation and a 0.6 mL methoprene formulation were drawn into an injection cylinder connected to the aerosol spray system. The flow rate into the sprayer was 1.8 mL/min for both particle sizes. The pressures employed to produce particles of VMDs 4 µm and 16 µm were 28 psi (193.0 kPa) and 7 psi (48.3 kPa), respectively. 

A total of 16 arenas were randomly placed on the grilled floor of the chamber with a clearance of 2.5 cm between arenas for exposure interval of 15 min for each spray treatment. In between each replicate spray treatment, the chamber was vented for 5 min to ensure that all aerosol particles were removed. The analytical equipment attached to the vertical flow chamber continually measured the aerosol particles inside the chamber, and after this 5 min venting period, the aerosol particles were below the limits of detectability with only background particle levels being present. Each of the arenas in the chamber during a replicate spray application was randomly assigned to each treatment combinations of weeks after aerosol treatment, temperatures and species. The spray process for each particle size was replicated 5 times (A total of 80 arenas were treated for each particle size and only 72 were used per particle size) and treated arenas were transported back to the CGAHR facility. 

The potential for degradation of activity due to temperature was explored by storing the treated arenas at different temperatures of 25, 30, 35, and 40 °C in four different incubators (Percival Scientific, Perry, IA, USA) (60% r.h and 16: 8 h L:D). In each incubator, a total of 36 treated arenas covered with lids were placed (3 replicate arenas for both particle sizes, 3 different periods for residual assessments for both species). As controls for aerosol treatments, a companion set of nine untreated arenas were also placed in each incubator for each species. Both the control arenas and treated arenas were haphazardly placed across the racks inside the incubators. To minimize positional effects within each chamber, arenas were switched between racks on weekly basis.

At 1 week after spray treatments, 18 arenas from each incubator representing all particle sizes (including controls) were removed from each incubator and placed in a walk-in growth chamber (KPS Global, Fort Worth, TX, USA) maintained at 30 °C, 60% r.h, 16:8h L:D. Arenas were removed one day before residual bioassays were initiated and this was required for conditioning the arenas to temperature in the walk-in growth chamber. In each treated and untreated arena, about 1 g of ground dog food was placed and arenas were shaken to spread the diet. Ten larvae of *T. inclusum* or *T. variabile* were introduced in each arena (size of larva ca. 4 mm). This larval exposure process was repeated at 3 weeks and 6 weeks after the treatments using new treated and untreated arenas. Development in the arenas was observed over a period of four weeks.

The residual efficacy of the pyrethrin + methoprene mixture was determined based on percentage of normal adult emergence and the calculation of a developmental index as used earlier in studies on this aerosol mixture [7,17]. The percentage of normal adult emergence was obtained by dividing the number of morphologically normal adults emerged with the total number of immatures released per arena and multiplying this fraction with 100. The developmental index in this study, as described earlier by Arthur et al. [7], captures the range of morphological deformities resulting from the residual exposure. The developmental index for each individual insect was scored as follows. Larvae that had not completed development beyond the larval stage, or develop as supernumerary larvae, or half larvae-half pupae intermediates, were scored as 1. Individuals that had not completed beyond the pupal stage or were half pupae–half adult intermediates were scored as 2. Emerged adults with major morphological deformities (deformed body parts, severely twisted wings, incomplete sclerotization) were scored as 3. Emerged adults with minor deformities (primarily twisted wings, but otherwise morphologically the same as untreated controls) were scored as 4, Morphologically normal adults were scored as 5. The arena developmental index was then determined by totaling the scores for all the individuals in each arena. For example, if all 10 larvae in an arena successfully emerged as morphologically normal adults, the arena developmental index would be 50. The developmental index per insect was then calculated by dividing the arena developmental index by the total number of larvae released in each arena. 

The impacts of particle size, temperature, species and residual week on percent normal adult emergence and developmental index per insect were analyzed by four-way ANOVA in GLIMMIX procedure using the Statistical Analysis System version 9.4 [18]. Experiment and experiment * particle size * species * residual week interactions were used as random terms. Finally, separate ANOVAs were conducted for each residual bioassay week to determine differential influence of aerosol treatments across both the species. Kenward-Roger’s denominator degrees of freedom method was used for appropriate degrees of freedom. For post hoc comparisons between treatments and interactions between treatments, Tukey–Kramer method of mean separation was used.

## 3. Results

The four-way ANOVA involving particle size, species, temperature and residual week revealed the overall effects of particle size and species to be significant on both adult emergence and developmental index; however, most interactions were not significant (Table 1). Only the species and particle size interaction were significant (Table 1). Temperature as a main effect was nonsignificant for both adult emergence and developmental index. 

Analysis of data for week 1 revealed similar results as in the overall four-way ANOVA, with no significant influence of temperature on either adult emergence (F = 0.12, df = 3,46, *p* = 0.9) or developmental index (F = 0.64, df = 3,46, *p* = 0.9). At week 1, the particle size significantly impacted both adult emergence and developmental index (adult emergence: *F* = 122.0, df = 2,46, *p* < 0.0001; developmental index: *F* = 147.7, df = 1,46, *p* < 0.0001). Also, both these variables were significantly influenced by species (adult emergence: *F* = 56.8, df = 1,46, *p* <0.0001; developmental index: *F* = 55.1, df = 1,46, *p* < 0.0001). Both particle sizes resulted in significant reduction in normal adult emergence in *T. variabile* compared to controls and the difference in adult emergence between both particle sizes was also significant (Table 2). In *T. inclusum*, although both particle sizes led to reduced adult emergences compared to the control, the difference in adult emergence between controls and the particle size of 4 µm was not significant and only the larger particle size of 16 µm resulted in a significant reduction compared to the other two treatments. The post-hoc comparison of developmental indices at week 1 between particle sizes for both species revealed similar patterns as in adult emergence with a reduction in developmental indices with increase in particle size. In contrast to *T. variabile* only the 16-µm particle size led to significant reduction in the developmental index values compared to control and 4-µm particle size in *T. inclusum* (Table 2). The interaction between particle size and species was evident by differential responses of *T. variabile* and *T. inclusum* to aerosol treatments with respect to both adult emergence (*F* = 13.2, df = 2,46, *p* < 0.0001) and developmental index (*F* = 24.0, df = 2,46, *p* < 0.0001) (Table 2). 

At residual week 3, adult emergence was influenced by particle size (*F* = 167.2, df = 2,46, *p* < 0.0001), species (*F* = 65.3, df = 1, 46, *p* < 0.0001) and interactions between particle size and species (*F* = 28.7, df = 2,46, *p* < 0.0001), but not by temperature (*F* = 0.34, df = 3,46, *p* = 0.8). Similarly, impact of treatments except for temperature were significant for developmental index (particle size: *F* = 252.6, df = 2,46, *p* < 0.0001; species: *F* = 123.8, df = 1,46, *p* < 0.0001, and interaction: *F* = 79.6, df = 2,46, *p* < 0.0001; temperature: *F* = 0.47, df = 3,46, *p* = 0.7). In *T. variabile*, adult emergence in both particle sizes was significantly reduced compared to the control and the lowest emergence at 16-µm particle size differed significantly from emergence at the 4-µm particle size (Table 3). In contrast, though the developmental index was reduced at both particle sizes, it was significantly reduced only at the 16-µm particle size. In *T. inclusum* only the 16-µm particle size significantly reduced both adult emergence and the developmental index. The differential impact of particle sizes between species was evident in greater magnitude for *T. variabile* compared to *T*. *inclusum* (Table 3). 

As in residual bioassays earlier at week 1 and week 3 post treatments, week 6 residual assays revealed significant effects of treatments on both adult emergence (particle size: *F* =163.6, df = 2,46, *p* < 0.0001; species: *F* = 48.0, df = 1,46, *p* < 0.0001; interaction: *F* = 26.3, df = 2,46, *p* < 0.0001) and developmental index (particle size: *F* = 138.1, df = 2,46, *p* < 0.0001; species: 46.4, df = 1,46, *p* < 0.0001; interaction: *F* = 34.1, df = 2,46, *p* < 0.0001). Temperature showed no significant influence on either adult emergence (*F* = 0.75, df = 3,46, *p* = 0.5) or developmental index (*F* = 0.35, df = 3,46, *p* = 0.8). For *T. variabile*, a significant reduction with respect to adult emergence and developmental index occurred at 16 µm-particle size but not at 4-µm particle size and this contrasted with effectiveness of both particle sizes in week 1 and week 3 residual assays (Table 4). The tendency for smaller particle sizes to deteriorate in their efficacy seemed to be more pronounced in *T. variabile* than in *T. inclusum*. As in the residual bioassays at earlier weeks, only the 16-µm particle size significantly reduced both adult emergence and developmental index in *T. inclusum*. Although the *post hoc* comparison on impact of 4-µm particle size between species on either of the variables did not reveal statistical significance, adult emergence and developmental index appeared lower in *T. variabile* compared to those in *T. inclusum* as in week 3 bioassay (Table 4). 

## 4. Discussion

The variation in responses of *T. variabile* and *T. inclusum* after exposure to the residual bioassays in the current study, and between *D. maculatus* and *T. variabile* in an earlier study [7], suggests differential susceptibility among dermestids. The same earlier study showed that this aerosol mixture dispensed at 16 µm prevented exposed *T. variabile* larvae from reaching the adult stage, but most of the exposed *D. maculatus* larvae emerged as normal adults. High efficacy at 16 µm particle size in this study is consistent with direct exposure studies using pyrthrin + methoprene directly applied to adults and larvae of *T. confusum* [7,9]. In a more recent study [8], the authors suggested that larger particle sizes were more likely to adhere to the body of the insect or the treated surface. Results from our study were also consistent with results from studies on khapra beetle, *Trogoderma granarium* (Everts) [19]. Ghimire et al. found low larval mortalities of *T. inclusum* and *T. variabile* on deltamethrin-treated and beta cyfluthrin-treated surfaces, whereas adults were more susceptible to both these insecticides [20]. 

An overall decline in persistence of methoprene activity was evident in present study due to the increase in adult emergence of exposed larvae and increase in developmental index values from 1 to 6 weeks. The reduced persistence of methoprene in the six-week testing period in this study contrasted with the Arthur et al. study that showed no decline in activity for eight-week test period at particle size of 2 µm against *T. confusum* [7]. These contrasting results could be attributed to differences in species employed, particle size of aerosol employed, and exposure times resulting in differences in the amount of material deposited or reflect unexplained variation between treatments. Additionally, temperature was not a significant factor in reducing duration of residual efficacy in the present study, although under different conditions temperature might be a more significant factor. Although activity of methoprene against *T. inclusum* tended to decline from week 1 to week 6, this decrease in activity was not substantial compared to the decline in activity of the 4-µm treatment against *T. variabile*. This apparent difference in persistence of activity between both species could be due to differences in susceptibility to the insecticide between the species. These differences could be due to species differences in physiology or due to differences in time in laboratory culture. The *T. variabile* colony had been in culture for more than 30 years, while the *T. inclusum* colony had been in culture for about 6 years. 

The lack of effect of temperature on persistence of methoprene in this study is consistent with results from a previous study by Wijayratne et al. [16]. The short duration of exposure of 6 weeks in the present study, contrasted with the prolonged duration of exposure for up to 24-weeks to similar temperatures used in this study by Wijayratne et al. [16]. This aspect of high persistence of activity of methoprene was also reported by these authors even when arenas were held at the temperature as high as 65 °C to simulate the use of heat treatments inside storage facilities for 48-h period [16].

In field applications aerosol particle size and concentration undergo variation depending on application system, distance from point of application and time after application, resulting in spatial heterogeneity in distribution and control efficacy of aerosols. Also, physical complexity of the interior of most food facilities may create more spatial variation in dosage resulting in non-uniform insect population suppression [21]. Species variation in susceptibility to methoprene in this and other studies may indicate that species composition may affect control efficacy when aerosols are applied in mills or food storage facilities. The decline in persistence of methoprene at smaller particle size highlights the need for attaining optimal distribution of particle sizes to maximize control efficacy of aerosols. 

## 5. Conclusions

The particle size at which aerosols are applied will affect immediate and residual efficacy, particularly when a combination such as pyrethrins and the IGR methoprene is used. Individual insect species, even those in the same genera, will vary in their response to aerosols. Residual efficacy can be achieved using an IGR, but efficacy will decline over time. In addition, aerosol efficacy can be assessed through a direct measure such as emergence of adults from exposed larvae, or a developmental index to capture development from larval to adult stage.

## Figures and Tables

**Table 1 insects-10-00142-t001:** Analyses of variance of percent adult emergence and developmental index of dermestids in concrete arenas after late larvae were released in arenas at 1, 3 and 6 weeks following the application of aerosol at different particle sizes and storage of arenas at different temperature treatments of 25, 30, 35, and 40 °C.

Source of Variation ^a^	df	Percent Adult Emergence		Developmental Index	
		F	*p*	F	*p*
Particle size	2142	372.78	<0.0001	317.24	<0.0001
Temperature	3142	0.28	0.84	0.28	0.84
Particle size * temperature	6142	0.15	0.98	0.32	0.93
Species	1142	147.50	<0.0001	121.54	<0.0001
Particle size * species	2142	86.40	<0.0001	42.88	<0.0001
Temperature * species	3142	0.13	0.94	0.35	0.79
Particle size * temperature * species	6142	0.23	0.97	0.31	0.93
Residual week	2142	4.0	0.02	3.81	0.02
Particle size * residual week	4142	1.20	0.32	1.45	0.22
Temperature * residual week	6142	0.40	0.89	0.27	0.95
Particle size * temperature * residual week	12,142	0.23	0.99	0.25	0.99
Species * residual week	2142	0.24	0.78	0.55	0.58
Particle size * species * residual week	4142	0.72	0.58	2.00	0.10
Temperature * species * residual week	6142	0.16	0.99	0.2	0.98
Particle size * temperature * species * residual week	12,142	0.13	0.99	0.15	0.99

^a^ Asterisk symbol represents interaction between treatments.

**Table 2 insects-10-00142-t002:** Percent adult emergence and developmental index of dermestids in pyrethrin + methoprene aerosol treated concrete arenas after larvae were released in arenas at week one following the application of aerosol and storage of arenas in incubators at different temperature treatments.

Variable ^a^	Species ^b^	Particle Size		
		Control	4 µm	16 µm
Adult emergence	*T.variabile*	92.0 ± 9.6 aA	43.3 ± 9.6 bB	2.7 ± 9.6 bC
	*T. inclusum*	93.1 ± 9.6 aA	80.8 ± 9.6 aA	45.3 ± 9.6 aB
Developmental index per insect	*T.variabile*	4.88 ± 0.25 aA	4.10 ± 0.25 bB	1.74 ± 0.25 bC
	*T.inclusum*	4.91 ± 0.25 aA	4.68 ± 0.25 aA	3.60 ± 0.25 aB

^a^ The LS Means on adult emergence and developmental index for each particle size by species combination followed by same lowercase letter indicate no significant difference between species for a given particle size and the means followed by same uppercase letter indicate no significant difference between particles for a given species at *p* < 0.05 (Tukey’s HSD). ^b^ Bioassays for both species were conducted at 30 °C with 3 replicate blocks (n = 12).

**Table 3 insects-10-00142-t003:** Percent adult emergence and developmental index of dermestids in pyrethrin + methoprene aerosol treated concrete arenas after late larvae were released in arenas at week three following the application of aerosol and storage of arenas in incubators at different temperature treatments.

Variable ^a^	Species ^b^	Particle Size		
		Control	4 µm	16 µm
Adult emergence	*T. variabile*	92.6 ± 4.6 aA	65.5 ± 4.6 bB	1.6 ± 4.6 bC
	*T. inclusum*	92.5 ± 4.6 aA	83.5 ± 4.6 aA	55.3 ± 4.6 aB
Developmental Index per insect	*T. variabile*	4.83 ± 0.11 aA	4.50 ± 0.11 aA	1.96 ± 0.11 bB
	*T. inclusum*	4.92 ± 0.11 aA	4.74 ± 0.11 aA	4.07 ± 0.11 aB

^a^ The LS Means on adult emergence and developmental index for each particle size by species combination followed by same lower-case letter indicate no significant difference between species for a given particle size and the means followed by same upper-case letter indicate no significant difference between particle sizes for a given species at *p* < 0.05 (Tukey’s HSD). ^b^ Bioassays for both species were conducted at 30° C with 3 replicate blocks (n = 12).

**Table 4 insects-10-00142-t004:** Percent adult emergence and developmental index of dermestids in pyrethrin + methoprene aerosol treated concrete arenas after late larvae were released in arenas at week six following the application of aerosol and storage of arenas in incubators at different temperature treatments.

Variable ^a^	Species ^b^	Particle Size		
		Control	4 µm	16 µm
Adult emergence	*T.variabile*	94.9 ± 5.2 aA	70.2 ± 5.2 aA	2.2 ± 5.2 bB
	*T. inclusum*	93.3 ± 5.2 aA	84.3 ± 5.2 aA	54.1 ± 5.2 aB
Developmental Index per insect	*T.variabile*	4.91 ± 0.18 aA	4.60 ± 0.18 aA	1.97 ± 0.11 bB
	*T.inclusum*	4.91 ± 0.18 aA	4.80 ± 0.18 aA	3.92 ± 0.11 aB

^a^ The LS Means on adult emergence and developmental index for each particle size by species combination followed by same lower case letter indicate no significant difference between species for a given particle size and the means followed by same upper case letter indicate no significant difference between particle sizes for a given species at *p* < 0.05 (Tukey’s HSD). ^b^ bioassays for both species were conducted at 30 °C with 3 replicate blocks (n = 12).

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
