# Peer review of "Evaluation of Residual Efficacy of Pyrethrin + Methoprene Aerosol on Two Dermestids: Impact of Particle Size, Species, and Temperature"

_insects, 2019, doi:10.3390/insects10050142_

Round 1

Reviewer 1 Report

This manuscript investigates “Evaluation of Residual Efficacy of Pyrethrin + 2 Methoprene Aerosol on Two Dermestids: Impact of Particle Size, Species and Temperature". This information is of interest to toxicologist and entomologists working within pesticides and merits publication. The experimental set up of this study appears to be well-designed and the data collected carefully. I think that this manuscript requires substantial rewriting to make its results clearer and more readily interpretable to the reader. The authors provide knowledge in this manuscript and can be of great interest to the journal. Based on the comments above reported, my opinion is that this manuscript may be suitable for printing on this journal.

A few points:

Line 27: keywords should be in alphabetic order

Line 31: Define “IPM”

Line 36: …indirect control of immature stages to exposure of residues? Why? This is unclear, rewrite

Line 40: delete “directly”

Line 40: "Jacqueline Du Val"...In manuscript, place the identifier name for all species

Line 52: …this IGR? You mean the methoprene?

Line 53: …T. variable (Dermestidae), are important pest that infest mills…  

Line 100: delete “Approximately”

Lines 115, 117: place ml by mL

Line 141: change “1.0” by “1”

Line 143: change “4.0” by “4”

Lines 145-151: The authors should provide the formula for this variable

Lines 249-251: Here you can start the discussion with the results obtained in this study and not mention preliminary works.

Line 268: change “tempature” by “temperature”

Line 279: change “etal.” by “ et al.”

Author Response

Reviewer 1’s comments and responses from authors

Line 27: keywords should be in alphabetic order

Our keywords are rearranged in alphabetical order as suggested (see line # 28).

Line 31: Define “IPM”

We spelled this out as “integrated pest management.” (line #s 32-33)

Line 36: …indirect control of immature stages to exposure of residues? Why? This is unclear, rewrite

Pyrethrins are expected to show their short-term immediate effects on exposed insects as they directly contact with aerosol being sprayed while IGR component provides long term residual control to exposed immatures. The sentence now reads as “Their combined use can accomplish insect population control by direct initial kill of exposed immatures and adults due to immediate contact toxicity of pyrethrin and delayed control of immature stages through exposure to residues of the IGR on treated surfaces.” (line # 37)

Line 40: delete “directly”

Deleted.

Line 40: "Jacqueline Du Val"...In manuscript, place the identifier name for all species

Have made sure that the names of authors are present at all first mentions of a species (line #46, 47 and 55).

Line 52: …this IGR? You mean the methoprene?

We changed to methoprene to make clearer.

Line 53: …T. variable (Dermestidae), are important pest that infest mills…  

Thanks for editing. Included Dermestidae in parenthesis. Now this sentence reads “Trogoderma inclusum LeConte, larger cabinet beetle and T. variabile (Coleoptera: Dermestidae), are important pests that infest mills, processing plants, and a wide range of grain products.

Line 100: delete “Approximately”

Deleted. The sentence in line # 102 now reads “A total of 720 arenas were prepared for this study.”

Lines 115, 117: place ml by mL

Changed ml to mL.

Line 141: change “1.0” by “1”

Changed.

Line 143: change “4.0” by “4”

Changed.

Lines 145-151: The authors should provide the formula for this variable

Given the simplicity of the formula for developmental index, we think an explanation in the text is enough. Have revised the text to make the calculation clearer (see line # 161-162 and # 163-165).

Lines 249-251: Here you can start the discussion with the results obtained in this study and not mention preliminary works.

We disagree with this change, since summarizing the results of the current and previous research goes toward the making of the overall conclusions presented.

Line 268: change “tempature” by “temperature”

Have made the correction.

Line 279: change “etal.” by “ et al.”

Have made the correction.

Reviewer 2 Report

This work is an extremely interesting research paper and has valuable information for the readers. Authors evaluated the residual efficacy of pyrethrin and methoprene mixture aerosol on T. variabile and T. inclusum. They looked at the effect of two aerosol particle sizes and 4 temperatures for storing treated arenas on residual efficacy by comparing adult emergence and developmental index. The study was well designed and performed. Statistical analysis is well done. I just have few comments and suggestions.

Specific comments are listed below.

1.      Page 3 line 128: Please indicate what CGAHR stands for.

2.      Page 3 line 133: please indicate the number of control arenas per incubator.

3.      Lines 141-142: where did you placed the food in the arena?

4.      Please mention the number of days that the experiment was running. In developmental index how did you determine that the larvae were unable to complete development beyond the larval stage?

5.      I would suggest to remove all the stats from the text and put it in one table similar to Table 1. Add a column for “residual week” or “week”. To separate the analysis for each week. That will make it easier to follow the results.

6.      Please put tables 2, 3 and 4 into one table. Again add a column for “residual week” or “week”. That would make it easier to look at the decline in persistence of methoprene and save space.

7.      Please add (n=36 for each species) in Tables 2, 3 and 4. 3treatments*3replicates*4temperature*2sp.=72

8.      Table 2 line 205: superscript “b” in “…P < 0.05 (Tukey’s HSD). b Bioassays…”

9.      Table 2, 3 and 4: Please add Developmental Index per Insect.  

10.  Page 6 line 227: Please be specific about the treatments in "significant effects of treatments on both".

11.  Page 7 lines 305-306: “Arthur, F.H., 2012. …, and food warehouses. J. Pest Sci. 2010, 85, 323-329.” Remove 2012 on line 305 and change 2010 to 2012 on line 306. Please check all your references.

12.  It would be interesting to determining the efficacy of pyrethrin and methoprene mixture aerosol on resistant populations.

Author Response

Reviewer 2’s comments and responses from authors:

1. Page 3 line 128: Please indicate what CGAHR stands for

We included acronym at the first mention of the name.

2.      Page 3 line 133: please indicate the number of control arenas per incubator.

In the latest version we included: “As controls for aerosol treatments, a companion set of nine untreated arenas were also placed in each incubator for each species.”

3.      Lines 141-142: where did you placed the food in the arena?

We have provided the details. Please see lines 142-143: “In each treated and untreated arena, about 1 g of ground dog food (~1 g) was placed and arenas were shaken to spread the diet.”

4.      Please mention the number of days that the experiment was running. In developmental index how did you determine that the larvae were unable to complete development beyond the larval stage?

Larvae that had not completed development beyond the larval stage within 4 weeks of bioassay period were considered arrested at that developmental stage.

Inserted this statement in line # 147. “Development in the arenas was observed over a period of four weeks.”

5.      I would suggest to remove all the stats from the text and put it in one table similar to Table 1. Add a column for “residual week” or “week”. To separate the analysis for each week. That will make it easier to follow the results.

We disagree with this change, since we think the text reads well with the statistical information inserted and putting all of this information into a single table will create a very large table that will take up a lot more space and be less user friendly in terms of evaluating the strength of statistical support for each of the specific results as read through the text.   

6.      Please put tables 2, 3 and 4 into one table. Again add a column for “residual week” or “week”. That would make it easier to look at the decline in persistence of methoprene and save space.

Rebuttal: This is a matter of style preference and we would like to retain separate tables. 

7.      Please add (n=36 for each species) in Tables 2, 3 and 4. 3treatments*3replicates*4temperature*2sp.=72

We stated in earlier version, for n as 36 for each species. We now revised this to n=12 as this size is appropriate to each droplet size per species. Please see the foot notes for table no 2, 3 and 4 for this change. 

8.      Table 2 line 205: superscript “b” in “…P < 0.05 (Tukey’s HSD). b Bioassays…”

Correction made.

9.      Table 2, 3 and 4: Please add Developmental Index per Insect.  

Added.

10.  Page 6 line 227: Please be specific about the treatments in "significant effects of treatments on both".

While presenting the results of statistical tests, treatments were mentioned in parenthesis (please see the line # 231-232.

11.  Page 7 lines 305-306: “Arthur, F.H., 2012. …, and food warehouses. J. Pest Sci2010, 85, 323-329.” Remove 2012 on line 305 and change 2010 to 2012 on line 306. Please check all your references.

Corrected.

12.  It would be interesting to determining the efficacy of pyrethrin and methoprene mixture aerosol on resistant populations.

This is an interesting research idea that could be followed up on with additional experiments.